# Temporal Trends in the Completeness of Epidemiological Variables in a Hospital-Based Cancer Registry of a Pediatric Oncology Center in Brazil

**DOI:** 10.3390/ijerph21020200

**Published:** 2024-02-09

**Authors:** Jonathan Grassi, Raphael Manhães Pessanha, Wesley Rocha Grippa, Larissa Soares Dell’Antonio, Cristiano Soares da Silva Dell’Antonio, Laure Faure, Jacqueline Clavel, Luís Carlos Lopes-Júnior

**Affiliations:** 1Graduate Program in Public Health, Universidade Federal do Espírito Santo (Ufes), Vitoria 29047-105, ES, Brazil; jonathangrassi17@hotmail.com (J.G.); rafa.09pessanha@gmail.com (R.M.P.); wgripa@gmail.com (W.R.G.); 2Espírito Santo State Health Department, Special Center for Epidemiological Surveillance, Vitoria 29047-105, ES, Brazil; lissadellantonio@gmail.com (L.S.D.); cristianosss@outlook.com (C.S.d.S.D.); 3Centre de Recherche en Epidémiologie et Statistiques (CRESS), Institut National de la Santé et de la Recherche Médicale—INSERM, Université Paris-Cité, 75013 Paris, France; laure.faure@inserm.fr (L.F.); jacqueline.clavel@inserm.fr (J.C.)

**Keywords:** hospital-based cancer registry, pediatric oncology, epidemiology, Brazil

## Abstract

This ecological time series study aimed to examine the temporal trends in the completeness of epidemiological variables from a hospital-based cancer registry (HbCR) of a reference center for pediatric oncology in Brazil from 2010 to 2016. Completeness categories were based on the percentage of missing data, with the categories excellent (<5%), good (5–10%), regular (11–20%), poor (21–50%), and very poor (>50%). Descriptive and bivariate analyses were performed using R.4.1.0; a Mann–Kendall trend test was performed to examine the temporal trends. Variables with the highest incompleteness included race/color (17.24% in 2016), level of education (51.40% in 2015), TNM (56.88% in 2012), disease status at the end of the first treatment (12.09% in 2013), cancer family history (79.12% in 2013), history of alcoholic consumption (39.25% in 2015), history of tobacco consumption (38.32% in 2015), and type of admission clinic (10.28% in 2015). Nevertheless, most variables achieved 100% completeness and were classified as excellent across the time series. A significant trend was observed for race/color, TNM, and history of tobacco consumption. While most variables maintained excellent completeness, the increasing incompleteness trend in race/color and decreasing trend in TNM underscore the importance of reliable and complete HbCRs for personalized cancer care, for planning public policies, and for conducting research on cancer control.

## 1. Introduction

Pediatric tumors account for 1 to 4% of all malignant tumors in most populations [1,2,3], with the proportion varying according to the population’s age distribution [3]. In low- and middle-income countries (LMICs), where the child population reaches about 50%, this proportion reaches between 3 and 10% of all neoplasms [4]. Pediatric cancer patients living in high-income countries (HICs) achieve very good outcomes, with approximately 80% surviving 5 years after diagnosis [5]. However, LMICs cover more than 90% of children and adolescents who are at risk of developing childhood cancer annually [4,6].

According to estimates for the triennium 2023–2025, Brazil is expected to have 7930 new cancer cases each year, which corresponds to an estimated risk of 134.81 per million children and adolescents (between 0 and 19 years old), and of these there is an estimated risk of 140.50 new cases per million male children and 128.87 per million female children [7].

Despite being regarded as one of the most significant achievements in modern science, the progress in the prognosis of children with cancer witnessed in HICs in the past few decades has not been replicated in most LMICs, where available data indicate that a much lower number of children survive [8,9,10,11]. Many of these countries lack accurate information on the incidence and outcomes of pediatric neoplasms. This is partly due to the absence of cancer registration and vital registration systems required to document and report data [3,11,12].

Hospital-based cancer registries (HbCRs) have been developed in many LMICs, particularly in Latin America, as well as Asia. In summary, they are very useful for multiple purposes such as providing information about the diagnosis and treatment of patients related to the specific tumor characteristics, as well as clinical outcomes [13].

In Brazil, the HbCR was established by the Brazilian National Cancer Institute (INCA) to enable the standardization of technical training and improve hospital management for cancer patients at the national level. The HbCR collects data on malignant neoplasm cases diagnosed as well as treated at a specific hospital [14] and serves as a valuable source of information for clinical and epidemiological research, including the evaluation of treatment outcomes and patient survival analysis using therapeutic protocols. Several studies have utilized HbCR data to improve patient care and inform cancer control policies [9,10,13,15,16,17,18,19].

The present study aimed to examine the temporal trends in the completeness of epidemiological variables from a HbCR of a reference center for pediatric oncology in Brazil.

## 2. Methods

### 2.1. Study Type

An ecological time series study was undertaken on the quality of epidemiological variables regarding the completeness of data available from a HbCR of a pediatric oncology center in a Brazilian state.

### 2.2. Eligibility

The definition of quality dimensions was used to assess the completeness of data in the hospital-based cancer registry (HbCR) [20]. Completeness refers to the proportion of fields filled with non-null values, and the proportion of missing data was classified using the classification proposed by Romero and Cunha (2006) [21], as described in detail in other previous publications [19,22]. In this study, a field filled with the category “ignored”, the numeral zero, an unknown date, or a term indicating missing data were considered to represent incompleteness [19,22].

We used secondary data from a single High Complexity Oncology Unity (UNACON) of the public network for treating cancer in children and adolescents (from 0 to 19 years old) in Espírito Santo state, Brazil, from 2010 up to 2016. Data were retrieved from the Secretaria de Estado da Saúde do Espírito Santo (SESA/ES).

### 2.3. Population and Data Collection

We utilized secondary data from Espírito Santo, a state in the Southeast Region of Brazil. The Espírito Santo Oncology Care Network operates in three health regions, namely, Metropolitan, South, and North/Central [19,22]. All cancer cases in children and adolescents in the state are attended to by a single UNACON located in the state capital, Vitória, at the Hospital Estadual Infantil Nossa Senhora da Glória (HEINSG). HEISNG has a well-established HbCR, and its databases are sent annually to the Brazilian Hospital Cancer Registry Integrator. The hospital also has an Oncology Care Line, which facilitates the care network’s flow in the state, improves accessibility to procedures related to cancer diagnosis and treatment, and enhances access to health services throughout the state [22,23].

The study population consisted of cancer cases in children and adolescents (0 to 19 years old) who had their first consultation at the hospital between 2010 and 2016. Only cases registered as analytical cases were included, in which the cancer registry hospital carried out the patient’s treatment planning, regardless of whether any stage of the treatment was conducted in another institution.

In this study, we have included hematopoietic malignancies and solid tumors, including the following: leukemias, central nervous system tumors, lymphomas, germ cell tumors, renal tumors, bone tumors, soft tissue sarcomas, sympathetic nervous system tumors, carcinomas, retinoblastomas, liver tumors, and other unspecified malignant tumors according to the International Classification of Childhood Cancer, 3rd edition (ICCC-3) and International Classification of Diseases for Oncology (ICD-O) [24,25].

Data were collected between September 2021 and December 2021 at the SESA/ES. We have opted the time frame of 2010 to 2016 because, until December 2016, the HEINSG had consolidated and submitted the data from the HbCR to the Cancer Epidemiological Surveillance of the state. Additionally, due to the COVID-19 pandemic, hospitals in the Espírito Santo state faced operational difficulties and delays in sending HbCR data to the Epidemiological Surveillance. Despite this, to maintain the standardization of the historical series and ensure the inclusion of consolidated data from this hospital, we decided to proceed with the chosen time period.

### 2.4. Variables

The variables contained in the tumor registration form of the Brazilian HbCR Integrator System (Sis-RHC) [14,26] were as follows: (1) sex, (2) date of birth, (3) age, (4) place of birth, (5) race/color, (6) level of education, (7) provenance (city of residence at the time of diagnosis), (8) date of first consultation at the hospital, (9) date of first tumor diagnosis, (10) previous diagnosis and treatment, (11) most important basis for tumor diagnosis, (12) location of primary tumor, (13) histologic type of primary tumor, (14) TNM, (15) clinical tumor staging, (16) hospital treatment initiation clinic, (17) start date of first tumor-specific treatment in hospital, (18) first treatment received in hospital, (19) status disease at the end of the first hospital treatment, (20) date of patient’s death, (21) death by cancer, (22) cancer family history, (23) history of alcohol consumption (of children and adolescents), (24) history of tobacco use (of children and adolescents) (25) origin of referral, (26) clinic at hospital admission, (27) probable location of primary tumor, (28) laterality of the tumor, (29) occurrence of more than one tumor, (30) cost of tumor diagnosis in the hospital, and (31) cost of tumor treatment in the hospital.

### 2.5. Statistical Analysis

To analyze the data, we used R version 4.2.1 (R Foundation for Statistical Computing, Vienna, Austria). The description of completeness was presented by the observed frequency, percentage, and their respective completeness ratings. Friedman’s test was used to compare score classifications between years. The Mann–Kendall trend test was used to assess whether there was a statistically significant temporal trend between the years evaluated. For all analysis, the alpha was fixed at 5%.

### 2.6. Ethical Aspects

Approval was obtained by the Institutional Review Board of the Universidade Federal do Espírito Santo (Reference: 3,831,617). We also obtained approval and authorization from the SESA/ES, for the collection of secondary data and restricted data related to this research.

## 3. Results

### 3.1. Incompleteness and Classification of Sociodemographic and Clinical Variables in the Hospital Estadual Infantil Nossa Senhora da Glória’s Hospital-Based Cancer Registry

A total of 740 cases of malignant pediatric neoplasms were recorded in children and adolescents (99, 108, 109, 91, 110, 107, and 116 in 2010, 2011, 2012, 2013, 2014, 2015, and 2016, respectively) (Figure 1).

The variables with the greatest incompleteness were race/color (17.24% in 2016), level of education (51.40% in 2015), TNM (56.88% in 2012), disease status at the end of the first treatment in the hospital (12.09% in 2013), family history of cancer (79.12% in 2013), history of alcoholic consumption (39.25% in 2015), history of tobacco consumption (38.32% in the year 2015), and the hospital admission clinic (10.28% in 2015).

Throughout the study period, several variables achieved 100% completeness, including sex, date of birth, age, origin, date of first hospital visit, date of first diagnosis, histological type of primary tumor, clinical staging of the tumor, the clinic at the beginning of treatment in the hospital, the date of initiation of the first tumor-specific treatment in the hospital, and the date of death of the patients (Table 1 and Table 2).

The classification of the epidemiological variables from the HEINSG’s HbCR did not show any significant difference when compared. This indicates that the classification remained similar across the period from 2010 to 2016 (Table 3).

### 3.2. Incompleteness Trends of Sociodemographic and Clinical Variables in the Hospital Estadual Infantil Nossa Senhora da Glória’s Hospital-Based Cancer Registry from 2010 to 2016

There was a statistically significant trend observed in the completeness of sociodemographic and clinical variables from 2010 to 2016. The variables that showed a significant trend were race/color, TNM, and history of tobacco consumption. The trend for race/color was an increase in incompleteness over the years. On the other hand, the incompleteness of TNM showed a downward trend over the years, while the incompleteness of the history of tobacco consumption increased. The incompleteness of the other variables remained constant over the years (Table 4).

Figure 2 shows the trends in the completeness of the TNM variables and history of tobacco consumption between 2010 and 2016 that were statistically significant over the period studied.

## 4. Discussion

Our results showed that many variables for understanding the health–disease process were classified as ‘excellent’; for example, sex and date of first diagnosis. However, some registries presented a high incompleteness rate for relevant variables, such as level of education, history of tobacco and alcohol consumption, TNM, and family history. These findings are in line with a study that examined the quality of information, verifying the completeness and consistency of the HbCR in a state located in the Central-West Region of Brazil, and have also found that TNM and schooling were among the most incomplete data variables [27].

The variable “sex” was classified as excellent over the whole time period of the study. It is important that this variable presents a high completeness rate, because gender predicts the incidence of several types of pediatric cancers [28]. The low interpretational subjectivity required to record this information may explain the results of this finding. Similar results were found in other studies with excellent completeness for the sex variable [27,29].

The completeness of the race/color variable worsened over time, with an increasing tendency towards non-completion. However, this variable was regularly recorded in the last two years of the study. It is important to note that the study of race/color goes beyond just biological differences and represents a complex variable that encompasses economic as well as cultural meanings, highlighting inequities in access to cancer care. These results are consistent with other research carried out in Brazil [19,27,29,30,31]. It is worth noting that incomplete and incorrect registration of this variable presents a challenge in accurately assessing the actual need for health promotion and disease prevention programs in vulnerable populations [31]. Additionally, the race/color variable is relevant in expanding discussions on social inclusion and social, individual, and political programmatic vulnerability [19,32].

Indeed, the race/color variable represents a set of sociocultural exposures that demonstrate the existing social inequity [31]. A retrospective cohort study conducted at a women’s health reference center in a public hospital in the State of São Paulo evaluated the completeness of variables in elderly women diagnosed with breast cancer. The study found a trend of whitening and whiteness, with a worsening completeness of the race/color variable, and an increasing trend of non-completeness of the data [31]. In contrast, the present study found that completeness for the same variable increased from 2010 to 2016.

Another study highlighted the challenges in collecting and interpreting data related to the race/color variable. The authors found that data collection was inadequate due to limitations in the classifications of newborns and deaths, as the ideal and recommended method for classification is through self-reporting. Incompleteness data for this variable were classified as stable in Espírito Santo and decreasing in the Southeast Region and the rest of Brazil [29].

In this study, we found that the variable ‘level of education’ had a poor score in most of the years investigated, which is consistent with other studies conducted in the same state, other HbCRs [19], and other Brazilian regions [27,30,31]. This variable is of great importance in determining a patient’s prognosis, and improving its completeness is of clinical and epidemiological relevance [19,33]. Additionally, studying the level of education allows us to indicate a patient’s socioeconomic situation, when accurate information on income is not available [19,33]. Furthermore, low levels of education may also be associated with late diagnosis, indicating that education level can hinder access to early diagnosis, treatment, and prognosis [27,31]. Data regarding education are not important only due to the late notification of cancer by the oncologist. Its relevance also extends to the subsequent implications of the absence of oncological knowledge. For instance, survivors of childhood cancer face a high risk of cardio-oncological complications in the future and, given this, it is imperative to carry out regular exams in adulthood [34].

Alcohol and tobacco consumption variables are challenging to fill in accurately, particularly when it comes to cancer in children and adolescents, as their measurement is subjective. As a result, there is a risk of bias in quantifying the use of these substances, or the field may be left incomplete [35]. The trend test for patterns of alcohol consumption and tobacco use did not reveal statistical significance; however, similarities are identified between both variables. During this study, there was no contact with the professionals responsible for records at the reference hospital in order to verify the reason for the lack of data completion. However, it is assumed that, given the nature of the population studied, composed of children and adolescents, the lack of information suggests that the consumption of these substances did not occur. A strategy to mitigate the incompleteness of this variable would be to adopt the marking ‘not applicable’ when, in fact, there was no consumption. Even in the adult population of the same state, the alcoholism variable presents a poor filling pattern, with up to 50.1% of data missing. For the smoking variable, the percentages of incompleteness also presented a poor classification in almost the entire period analyzed [19].

A recent study, which aimed to investigate temporal trends and factors associated with cancers diagnosed at stage IV in Brazil over two decades, showed that there was a high frequency of cancer diagnosed at stage IV and an increasing trend in different cancer types, which were associated with distinct sociodemographic, lifestyle, and clinical factors [36].

Very poor completeness was verified for the TNM variable. Studies in Brazil [19,33] found a “poor” degree of completeness when analyzing the HbCR data. Conversely, excellent completeness was observed for the TNM variable in another state [31]. TNM is used worldwide, and it is extremely important to know about the disease [19,27,37,38]. The classification of malignant tumors (TNM) represents the main tumor classification system; thus, it is an essential variable in primary studies using databases. Therefore, knowing the staging is essential, making it possible to differentiate the tumor’s dimensions at diagnosis [27]. Additionally, knowing the tumor and its staging contributes to therapeutic planning and effective treatment. A recent Brazilian study reported an “excellent” and “good” rating in the database for the countryside of the state of Mato Grosso, Brazil, and a “poor” and “very poor” completeness in the database for the capital Cuiabá. In the present study, it was found that the percentage of completeness for the TNM variable decreased from 2010 to 2016 [27].

Because of their heterogeneity and rarity, childhood cancers already represent a particular data management challenge for registries. Hence, staging systems should be able to be applied by registry staff using available records and should be sufficiently detailed for the analysis and interpretation of population cancer data, while respecting the different capacities and resources of different registries [17]. Because most pediatric cancers have specific staging systems, general adult stage classifications are not appropriate.

Therefore, it is recommended that the tiered, pediatric-specific staging systems endorsed by the Toronto Pediatric Cancer Stage guidelines be adopted for pediatric cases by cancer registries in countries of all income levels and integrated into registry manuals. In addition, tiered staging systems should be endorsed, with more detailed systems for well-resourced cancer registries with appropriate data access, and less detailed systems for registries with limited resources and access. Lower-tier systems should be based on collapsing higher-tier system categories to preserve comparability across registries [17]. A family history (FH) of cancer is a hallmark for early detection of cancer [30,39,40,41]. Nevertheless, the FH variable in our study was classified as very poor in almost all periods studied. Family history is crucial to differentiate between cases of sporadic malignant neoplasms, those with family grouping, and those that are hereditary [38,39,40]. Identifying family members at risk for hereditary neoplastic syndromes is pivotal for preventive care, reducing cancer-related morbimortality and costs for health systems [42,43,44]. In comparison with another previous study from the same Brazilian state, but in the adult population, the family history of cancer variable also obtained a very poor classification in all years, with percentages of non-completion between 62.5% and 70.3%; the year with the highest amount of missing data was 2015, with 5980 (70.3%) [19].

Indeed, political commitments to report on cancer and to achieve better health outcomes in cancer care have accentuated the need for high-quality data, mainly taking into account the personalized care expected in the current precision medicine era [45,46]. However, only one in three countries has high-quality incidence data worldwide, and one in four has high-quality mortality statistics [11,45,46]. For instance, for childhood cancer at the age of 0–19 years, the acuity of the problem is particularly pronounced [45].

The present study has some limitations. Firstly, the study was conducted at a single pediatric oncology hospital in a state in the Southeast Region of Brazil, which may limit the generalizability of the findings to other settings. Secondly, all the analyses reported are based upon secondary cancer data, and, as we know, inherent to all health information systems is the issue of missing and incomplete data. Thirdly, while the hospital-based cancer registry (HbCR) provides valuable information about the quality of care provided, it may not reflect the entire regional or national cancer epidemiology. Nonetheless, this research addresses a topic of increasing epidemiological relevance and might help elucidate gaps in HbCRs across the world, mainly in middle-income-countries. Finally, the present study draws attention to the importance and relevance of complete and reliable data in HbCRs. When data follows the principles of accuracy, completeness, conformity, punctuality, consistency, and integrity, some actions become even more efficient; for example, accurate and reliable decision-making by healthcare professionals and clinicians can be guaranteed by the good management of data quality.

## 5. Conclusions

In conclusion, the completeness of most variables evaluated in the HEINSG’s HbCR in Espírito Santo, Brazil was rated as excellent. However, the variable race/color showed an increasing trend of incompleteness, while TNM displayed a downward trend between 2010 and 2016. We hope to continue this research, analyzing the subsequent time series (2017–2023) as soon as the data are available, aiming to compare the trends in completeness of the variables present in the tumor file for better monitoring. To better understand the health–disease process, high-quality data in HbCRs are necessary. A strategy to mitigate the occurrence of unfilled information would be the implementation of continuous training for registrars, highlighting the relevance of these registries as a basis for planning care, monitoring patients, and the quality of care provided in hospital institutions, as well as for conducting research focused on cancer surveillance.

## Figures and Tables

**Figure 1 ijerph-21-00200-f001:**
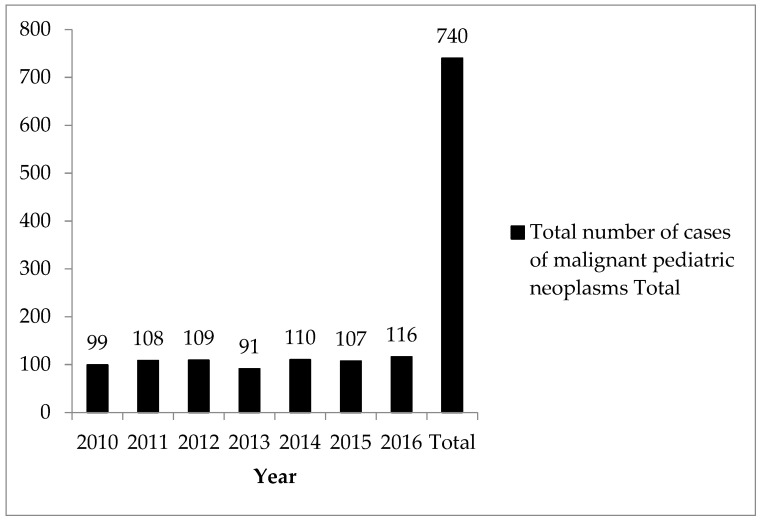
Total number of cases of malignant pediatric neoplasms in the registry.

**Figure 2 ijerph-21-00200-f002:**
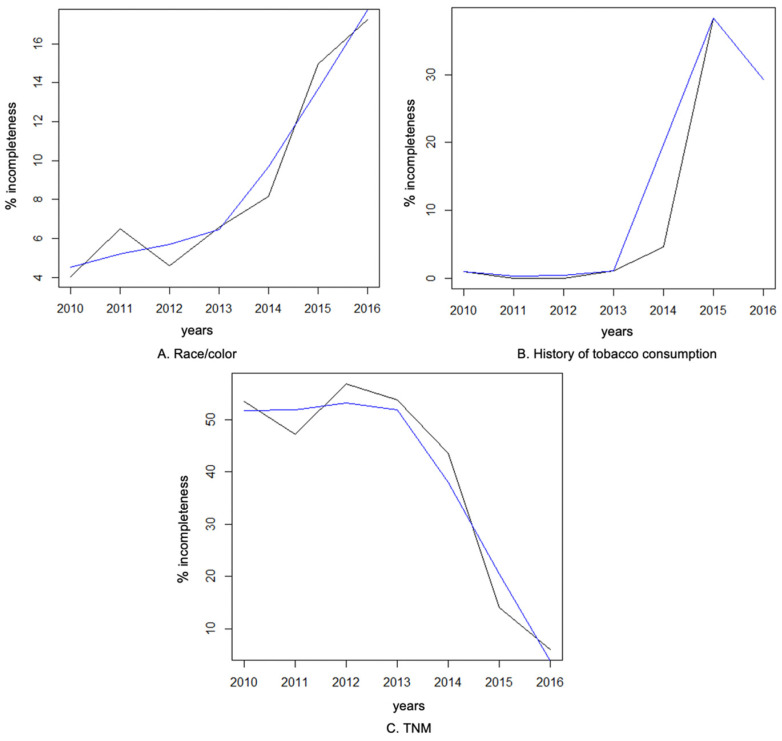
Trends of incompleteness of TNM variables and history of tobacco consumption in the HEINSG’s HbCR. Black lines represent the actual data and the blue lines represent trends.

**Table 1 ijerph-21-00200-t001:** Percentage of incompleteness and completeness classification for sociodemographic variables in the HEINSG’s HbCR.

Variables		2010	2011	2012	2013	2014	2015	2016
Sex	% Incompleteness	0.00	0.00	0.00	0.00	0.00	0.00	0.00
	Classification	Excellent	Excellent	Excellent	Excellent	Excellent	Excellent	Excellent
Birth Date	% Incompleteness	0.00	0.00	0.00	0.00	0.00	0.00	0.00
	Classification	Excellent	Excellent	Excellent	Excellent	Excellent	Excellent	Excellent
Age	% Incompleteness	0.00	0.00	0.00	0.00	0.00	0.00	0.00
	Classification	Excellent	Excellent	Excellent	Excellent	Excellent	Excellent	Excellent
Birthplace	% Incompleteness	1.01	0.00	0.00	0.00	0.00	0.93	0.00
	Classification	Excellent	Excellent	Excellent	Excellent	Excellent	Excellent	Excellent
Race/color	% Incompleteness	4.04	6.48	4.59	6.59	8.18	14.95	17.24
	Classification	Excellent	Good	Excellent	Good	Good	Regular	Regular
Level of education	% Incompleteness	40.40	39.81	42.20	27.47	8.18	51.40	47.41
	Classification	Poor	Poor	Poor	Poor	Good	Very Poor	Poor
Provenance	% Incompleteness	0.00	0.00	0.00	0.00	0.00	0.00	0.00
	Classification	Excellent	Excellent	Excellent	Excellent	Excellent	Excellent	Excellent
History of alcohol consumption	% Incompleteness	1.01	0.00	0.00	0.00	4.55	39.25	30.17
	Classification	Excellent	Excellent	Excellent	Excellent	Excellent	Poor	Poor
History of tobacco consumption	% Incompleteness	1.01	0.00	0.00	1.10	4.55	38.32	29.31
	Classification	Excellent	Excellent	Excellent	Excellent	Excellent	Poor	Poor
Cost of diagnosis	% Incompleteness	1.01	0.00	0.00	0.00	0.00	0.00	0.00
	Classification	Excellent	Excellent	Excellent	Excellent	Excellent	Excellent	Excellent
Treatment cost	% Incompleteness	1.01	0.00	0.00	0.00	0.00	0.93	0.00
	Classification	Excellent	Excellent	Excellent	Excellent	Excellent	Excellent	Excellent

**Table 2 ijerph-21-00200-t002:** Incompleteness percentages and completeness classification for diagnosis- and treatment-related variables in the HEINSG’s HbCR.

Variables		2010	2011	2012	2013	2014	2015	2016
First consultation date	% Incompleteness	0.00	0.00	0.00	0.00	0.00	0.00	0.00
	Classification	Excellent	Excellent	Excellent	Excellent	Excellent	Excellent	Excellent
Date of first diagnosis of the tumor	% Incompleteness	0.00	0.00	0.00	0.00	0.00	0.00	0.00
	Classification	Excellent	Excellent	Excellent	Excellent	Excellent	Excellent	Excellent
Previous diagnosis and treatment	% Incompleteness	0.00	0.00	0.00	0.00	0.00	1.87	0.00
	Classification	Excellent	Excellent	Excellent	Excellent	Excellent	Excellent	Excellent
Most important basis for diagnosis	% Incompleteness	0.00	0.00	0.00	0.00	0.00	0.93	0.00
	Classification	Excellent	Excellent	Excellent	Excellent	Excellent	Excellent	Excellent
Primary tumor location	% Incompleteness	0.00	0.00	1.83	0.00	0.00	0.00	0.86
	Classification	Excellent	Excellent	Excellent	Excellent	Excellent	Excellent	Excellent
Histological type of tumor first	% Incompleteness	0.00	0.00	0.00	0.00	0.00	0.00	0.00
	Classification	Excellent	Excellent	Excellent	Excellent	Excellent	Excellent	Excellent
TNM	% Incompleteness	53.54	47.22	56.88	53.85	43.64	14.02	6.03
	Classification	Very Poor	Poor	Very Poor	Very Poor	Poor	Regular	Good
Clinical staging of the tumor	% Incompleteness	0.00	0.00	0.00	0.00	0.00	0.00	0.00
	Classification	Excellent	Excellent	Excellent	Excellent	Excellent	Excellent	Excellent
Hospital treatment initiation clinic	% Incompleteness	0.00	0.00	0.00	0.00	0.00	0.00	0.00
	Classification	Excellent	Excellent	Excellent	Excellent	Excellent	Excellent	Excellent
First date specific treatment for the tumor	% Incompleteness	0.00	0.00	0.00	0.00	0.00	0.00	0.00
	Classification	Excellent	Excellent	Excellent	Excellent	Excellent	Excellent	Excellent
First treatment received at the hospital	% Incompleteness	0.00	0.00	0.00	0.00	0.00	0.00	0.86
	Classification	Excellent	Excellent	Excellent	Excellent	Excellent	Excellent	Excellent
Disease status at the end of the 1st treatment	% Incompleteness	0.00	0.00	0.92	12.09	11.82	3.74	5.17
	Classification	Excellent	Excellent	Excellent	Regular	Regular	Excellent	Good
Patient’s death date	% Incompleteness	0.00	0.00	0.00	0.00	0.00	0.00	0.00
	Classification	Excellent	Excellent	Excellent	Excellent	Excellent	Excellent	Excellent
Death by cancer	% Incompleteness	0.00	0.93	0.92	0.00	0.00	2.80	1.72
	Classification	Excellent	Excellent	Excellent	Excellent	Excellent	Excellent	Excellent
	Classification	Excellent	Excellent	Excellent	Excellent	Excellent	Excellent	Excellent
Family history of cancer	% Incompleteness	46.46	51.85	65.14	79.12	76.36	65.42	37.07
	Classification	Poor	Very Poor	Very Poor	Very Poor	Very Poor	Very Poor	Poor
Origin of referral	% Incompleteness	1.01	0.00	0.00	0.00	0.91	2.80	0.86
	Classification	Excellent	Excellent	Excellent	Excellent	Excellent	Excellent	Excellent
Clinic at hospital admission	% Incompleteness	1.01	0.00	0.00	0.00	0.91	10.28	1.72
	Classification	Excellent	Excellent	Excellent	Excellent	Excellent	Regular	Excellent
Probable location of tumor first	% Incompleteness	0.00	0.00	1.83	0.00	0.00	0.00	0.00
	Classification	Excellent	Excellent	Excellent	Excellent	Excellent	Excellent	Excellent
Tumor laterality	% Incompleteness	2.02	0.00	0.00	0.00	0.00	4.67	3.45
	Classification	Excellent	Excellent	Excellent	Excellent	Excellent	Excellent	Excellent

**Table 3 ijerph-21-00200-t003:** Comparison of completeness classification scores.

Year	Middle Rank	*p*-Value *
2010	4.21	0.086
2011	4.11	
2012	4.11	
2013	3.93	
2014	4.09	
2015	3.60	
2016	3.94	

(*) Friedman test; significant if *p* < 0.050.

**Table 4 ijerph-21-00200-t004:** Trend analysis of incompleteness for selected epidemiological variables in the HEINSG’s HbCR.

Variables	S	*p*-Value *	Trend
Sex	-	-	-
Birth Date	-	-	-
Age	-	-	-
Birthplace	−3.0	0.704	NS
Race/color	19.0	0.004	Increase
Level of education	3.0	0.764	NS
Provenance	-	-	-
First consultation date	-	-	-
Date of first diagnosis of the tumor	-	-	-
Previous diagnosis and treatment	4.0	0.454	NS
Most important basis for diagnosis	4.0	0.454	NS
Primary tumor location	3.0	0.704	NS
Histological type of tumor first	-	-	-
TNM	−13.0	0.003	Decrease
Clinical staging of the tumor	-	-	-
Hospital treatment initiation clinic	-	-	-
First date specific treatment for the tumor	-	-	-
First treatment received at the hospital	6.0	0.211	NS
Disease status at the end of the first treatment	10.0	0.172	NS
Patient’s death date	-	-	-
Death by cancer	6.0	0.433	NS
Family history of cancer	3.0	0.764	NS
History of alcohol consumption	10.0	0.158	NS
History of tobacco consumption	14.0	0.048	Increase
Forwarding origin	4.0	0.638	NS
Clinic at hospital admission	8.0	0.272	NS
Probable location of tumor first	−2.0	0.803	NS
Tumor laterality	5.0	0.503	NS
Occurrence of more than one tumor	−6.0	0.211	NS
Cost of diagnosis	−6.0	0.211	NS
Treatment cost	−3.0	0.704	NS

Abbreviations: TNM: tumor–node–metastasis; NS: not significant; (*): Mann–Kendall test; S: trend statistics, which were considered significant if *p* < 0.050. (-): Statistics not computed because completeness was 100.0%.

## Data Availability

Data are contained within the article.

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
