# Peer review of "Temporal Trends in the Completeness of Epidemiological Variables in a Hospital-Based Cancer Registry of a Pediatric Oncology Center in Brazil"

_ijerph, 2024, doi:10.3390/ijerph21020200_

Round 1
Reviewer 1 Report
Comments and Suggestions for Authors
The abstract must be reduced to 200 words.
References 25, 26 can be removed because it is a general method, not a method developed by someone in particular.
Line 135: adolescents (99, 108, 109, 91, 110, 107, 116 in 2010, 2011, 2012, 2013, 2014, 2015, and 2016, respectively). – These data look better in a graph.
The term "incompleteness" must be defined in the material and method.
Regarding the trends - there is a need for the gross classification of the data. For example, in terms of race - what exactly grows?
Regarding the TNM, it is a scoring system made up of 3 elements that do not always go together in terms of the trend. What does the T, the N or the M drop?
It should be specified or made a graph in relation to the types of cancer developed.
Does the consumption of tobacco or alcohol refer to parents or children? - must be mentioned.
What are the causes that lead to the lack of completion? Some aspects were touched upon, without going into detail.
Beware of self-citations. At least 20 references are not current.
In conclusion, the article is slightly superficial in certain aspects, it does not go into the essence of the problem.
Comments on the Quality of English LanguageCertain sentences need to be shortened. Certain words are used with a different meaning.
Author Response
REVIEWER 1
The abstract must be reduced to 200 words.
Response: Done !
References 25, 26 can be removed because it is a general method, not a method developed by someone in particular.
Response: Done! References were removed.
Line 135: adolescents (99, 108, 109, 91, 110, 107, 116 in 2010, 2011, 2012, 2013, 2014, 2015, and 2016, respectively). – These data look better in a graph.
Response: Thank you so much for this suggestion. We have insert a graphic as per recommended.
The term "incompleteness" must be defined in the material and method.
Response: We have added the definition in the methods as per recommended. Thank you so much.
Completeness refers to the extent to which the analyzed fields are filled, measured by the proportion of notifications with a category other than those indicating missing data. In this study, a field filled with the category “ignored,” the numeral zero, an unknown date, or a term indicating missing data were considered incomplete.
Regarding the trends - there is a need for the gross classification of the data. For example, in terms of race - what exactly grows?
Response: Over the years (from 2010 to 2016) the incomplete rate for this variable increased. In other words, in 2010 it was a variable with excellent completeness and over the years it changed to regular (there was little completion and/or little attention when recording this variable in recent years)
Regarding the TNM, it is a scoring system made up of 3 elements that do not always go together in terms of the trend. What does the T, the N or the M drop?
Response: Here we considered whether the TNM - stage variable was fulfilled or not. This is a variable that for most of the historical series remained highly incomplete.
It should be specified or made a graph in relation to the types of cancer developed.
Response: Thank you for this suggestion. We have added one paragraph addressing this issue.
The study included hematopoietic malignancies and solid tumors, such as: leukemias, Central Nervous System tumors, lymphomas, germ cell tumors, renal tumors, bone tumors, soft tissue sarcomas, sympathetic nervous system tumors, carcinomas, retinoblastomas , liver tumors, and other unspecified malignant tumors according to the International Classification of Childhood Cancer, 3rd edition (ICCC-3) and International Classification of Diseases for Oncology (ICD-O) [24,25].
Does the consumption of tobacco or alcohol refer to parents or children? - must be mentioned.
Response: We have added this information in methods. Thank you !
What are the causes that lead to the lack of completion? Some aspects were touched upon, without going into detail.
Response: the lack of preparation of registrars, lack of continued education to raise awareness among professionals about the importance of carefully and correctly filling out the HbCR. Possibly high turnover of professionals in the service, there may be some aspects that led to the lack of completeness of some variables.
Beware of self-citations. At least 20 references are not current.
Response: We carefully reviewed the manuscript that had 5 self-citations and now there are only 3 that are essential. Furthermore, we also update references as requested.
Reviewer 2 Report
Comments and Suggestions for Authors
Authors have written a research article entitled “Temporal Trends in the Completeness of Epidemiological Variables in a Hospital-Based Cancer Registry of a Pediatric Oncology Center in Brazil”. The theme is interesting; however, the manuscript is not properly discussed in context to results and data/experimental findings was not properly represented that the authors should clarify before a new peer-revision round. The study design of the manuscript is not impressive and does not add some more information related to the impact of this kind of study on Pediatric Oncology. Therefore, the manuscript is not suitable for publication; it should be carefully revised before the following peer-review process.
· What are the most recent and important achievements in the field related to present work? In my opinion, answers to these questions should be emphasized.
· Authors have not explained the rationale of the study to a greater extent. How could this study enhance the understanding/knowledge of researcher or clinicians by using this type of data.
· Results presented in this study are very limited, all the experimental analysis based upon secondary data of cancer. These data could not lead to the conclusion of present study. Some more parameters must be added in this study. Please use reference: https://doi.org/10.1016/j.canep.2022.102242
· There is poor presentation of data like very less use of graph and figure.
· The discussion should be rather organized around arguments avoiding simply describing details without providing much meaning. Authors have discussed some content and then seems not interested in writing more content which makes discussion part incomplete.
The whole manuscript needs to be checked by native English speakers for possible Grammatical corrections.
Comments on the Quality of English LanguageThe whole manuscript needs to be checked by native English speakers for possible Grammatical corrections.
Author Response
REVIEWER 2
Authors have written a research article entitled “Temporal Trends in the Completeness of Epidemiological Variables in a Hospital-Based Cancer Registry of a Pediatric Oncology Center in Brazil”. The theme is interesting; however, the manuscript is not properly discussed in context to results and data/experimental findings was not properly represented that the authors should clarify before a new peer-revision round. The study design of the manuscript is not impressive and does not add some more information related to the impact of this kind of study on Pediatric Oncology. Therefore, the manuscript is not suitable for publication; it should be carefully revised before the following peer-review process.
Response: This statement of yours sounds somewhat contradictory. We believe, as you said at the beginning of your comment, that The theme is interesting. Furthermore, this is a pioneering and unprecedented study in the only specialized and reference center in Pediatric Oncology in the State of Espírito Santo, Brazil. The Discussion was expanded according to pertinent notes from the other 3 reviewers.
What are the most recent and important achievements in the field related to present work? In my opinion, answers to these questions should be emphasized.Authors have not explained the rationale of the study to a greater extent. How could this study enhance the understanding/knowledge of researcher or clinicians by using this type of data.
Response: Added in the discussion as well as conclusion.
- Results presented in this study are very limited, all the experimental analysis based upon secondary data of cancer. These data could not lead to the conclusion of present study. Some more parameters must be added in this study. Please use reference: https://doi.org/10.1016/j.canep.2022.102242
Response: We have added this reference as per recommended. Thanks.
- There is poor presentation of data like very less use of graph and figure.
Response: We have added one more Figure.
- The discussion should be rather organized around arguments avoiding simply describing details without providing much meaning. Authors have discussed some content and then seems not interested in writing more content which makes discussion part incomplete.
Response: Thank you for this comment. The Discussion was expanded according to pertinent notes from the other 3 reviewers who endorsed our manuscript.
Reviewer 3 Report
Comments and Suggestions for Authors
The manuscript presents a detailed account of data completeness in the hospital-based cancer registry of a paediatric oncology centre over a 7-year period. It shows clearly the strengths and weaknesses in data acquisition over the study period, but would benefit from a little more searching commentary. I would ask the authors to consider the following points in revising their manuscript.
1 Introduction
Lines 42-45. Please specify age range for these estimates.
2.4 Variables
Line 109. Please explain the meaning of variable 7 (Provenance).
3 Results
Are completeness data available for variables 23 (Alcohol consumption) and 24 (Tobacco use) by age? It would be interesting to know if they differed markedly between ages 0-14 and 15-19.
Table 1. Please change the wording ‘Forwarding origin’ to ‘Origin of referral’ (as in the text at line 116), whose meaning is more readily understood.
Figure 1. The blue lines need to be explained in the Figure title or in a footnote. In graph B, the data point for 2016 is missing.
4 Discussion
Lines 223-226. The trend test for alcohol consumption is indeed not formally significant, but in Table 1 the patterns for alcohol consumption and tobacco use are strikingly similar, with incompleteness <=1.1% in 2010-2013, 4.55% in 2014, 38-39% in 2015 and 29-30% in 2016. This should be highlighted in the Discussion. Was there a conscious move towards leaving these variables blank or entering ‘None’ in the later years?
Reference 17 concerns Toronto staging, but there appears to be no further mention in the manuscript. A useful addition would be a brief discussion of the degree of completeness of data required for assigning Toronto stage and how this could be improved.
A comparison of the incompleteness levels and time trends between Tables 1 and 2 of the present manuscript and Tables 2 and 3 of reference 19 (which covers all HbCRs in the state and therefore includes predominantly adult patients) would be of interest.
Comments on the Quality of English LanguageEnglish language generally very good, but please attend to relevant points in Comments and Suggestions for Authors
Author Response
REVIEWER 3
The manuscript presents a detailed account of data completeness in the hospital-based cancer registry of a paediatric oncology centre over a 7-year period. It shows clearly the strengths and weaknesses in data acquisition over the study period, but would benefit from a little more searching commentary. I would ask the authors to consider the following points in revising their manuscript.
Response: Thank you so much for your positive feedback. We have addressed all the issues as per suggested.
Lines 42-45. Please specify age range for these estimates.
Response: Information added (between 0 and 19 years old),
Line 109. Please explain the meaning of variable 7 (Provenance).
Response: The meaning of the variable has been included. (city of residence at the time of diagnosis)
Are completeness data available for variables 23 (Alcohol consumption) and 24 (Tobacco use) by age? It would be interesting to know if they differed markedly between ages 0-14 and 15-19.
Response: Thank you for this valuable comment. In fact, this is a very timely and interesting comment, which we will address this issue in another manuscript. Here the objective is to show the completeness of the data in this historical series and we do not have stratification by age in this completeness analyses.
Table 1. Please change the wording ‘Forwarding origin’ to ‘Origin of referral’ (as in the text at line 116), whose meaning is more readily understood.
Response: Done. The term has been modified as per recommended.
Figure 1. The blue lines need to be explained in the Figure title or in a footnote. In graph B, the data point for 2016 is missing.
Response: OK. Done. It was included in the figure title: « Black lines represent the actual data and the blue lines the trends »
Lines 223-226. The trend test for alcohol consumption is indeed not formally significant, but in Table 1 the patterns for alcohol consumption and tobacco use are strikingly similar, with incompleteness <=1.1% in 2010-2013, 4.55% in 2014, 38-39% in 2015 and 29-30% in 2016. This should be highlighted in the Discussion. Was there a conscious move towards leaving these variables blank or entering ‘None’ in the later years?
Response: We have included the following paragraph in discussion as per recommended. Thanks !
The trend test for patterns of alcohol consumption and tobacco use did not reveal statistical significance; however, similarities are identified between both variables. During this study, there was no contact with those professionals responsible for records at the reference hospital in order to verify the reason for the lack of data completion. However, it is assumed that, given the nature of the population studied, composed of children and adolescents, the lack of information suggests that the consumption of these substances did not occur. A strategy to mitigate the incompleteness of this variable would be to adopt the marking 'not applicable' when, in fact, there was no consumption.
Reference 17 concerns Toronto staging, but there appears to be no further mention in the manuscript. A useful addition would be a brief discussion of the degree of completeness of data required for assigning Toronto stage and how this could be improved.
Response: We have added two paragraph into Discussion to addressing it. Thank you for this kind suggestion.
A comparison of the incompleteness levels and time trends between Tables 1 and 2 of the present manuscript and Tables 2 and 3 of reference 19 (which covers all HbCRs in the state and therefore includes predominantly adult patients) would be of interest.
Response: Thank you so much for this valuiable comment. We have included two paragraphs addressing this comparisons as per suggested. Many Thanks!
Even in the adult population of the same state, the alcoholism variable presents a poor filling pattern, with up to 50.1% of data missing. For the smoking variable, the percentages of incompleteness also presented a poor classification in almost the entire period analyzed [19].
In comparison with another previous study from the same Brazilian state, but in the adult population, the family history of cancer variable also obtained a very poor classification in all years, with percentages of non-completion between 62.5% and 70.3%; The year with the highest amount of missing data was 2015, with 5,980 (70.3%) [19].
Reviewer 4 Report
Comments and Suggestions for Authors
1. Very interesting work. It touches on an extremely important topic. The data missing from this registry could have a key impact on future education and treatment effectiveness.
2. The present study has some limitations. Firstly, the study was conducted at a single 252 pediatric oncology hospital in a state in the Southeast Region of Brazil, which may limit 253 the generalizability of the findings to other settings. Secondly, while the Hospital Cancer 254 Registry (HbCR) provides valuable information about the quality of care provided, it may 255 not reflect the entire regional, or national cancer epidemiology. Nonetheless, this research 256 addresses a topic of increasing epidemiological relevance and might help elucidate gaps 257 in HbCRs across the world, mainly, in middle-income-countries
A fair assessment of the weaknesses of the study-population from one region and one center.
Suggests planning a survey that takes into account:
1. Very importantly, the % of data such as race, education level, family history of cancer and the impact of the first line of treatment is decreasing with subsequent years.
The data ends with 2016-I suggest doing another similar work with current data 2017-2023 to see if it continues to look worse and worse over the years.
will take into account data from a larger part of the country
3. Additionally, studying the level of education al- 218 lows us to indicate socioeconomic situations when accurate information on income is not 219 available [19,33]. Furthermore, low levels of education may also be associated with late 220 diagnosis, indicating that education level can hinder access to early diagnosis, treatment, 221 and prognosis [27,31].
Please add that the education data is not only very important because of the late reporting of cancer to the doctor. They are also important because of the later consequences of lack of cardiooncologic knowledge: survivors of childhood cancer are at high risk of oncologic background complications in the future. Hence the need for regular examinations in adulthood. Please quote:
A practical approach to the 2022 ESC cardio-oncology guidelines: Comments by a team of experts - cardiologists and oncologists.
Kardiol Pol. 2023;81(10):1047-1063. doi: 10.33963/v.kp.96840. Epub 2023 Sep 3
4. Add to the conclusions that the proposal to improve the number of data. Perhaps if the assignment of appropriate treatment depended on the completion of this data, the data would be more complete?
Author Response
REVIEWER 4
Very interesting work. It touches on an extremely important topic. The data missing from this registry could have a key impact on future education and treatment effectiveness.
Response: Thank you so much for your comments as well as positive feedback regarding our paper.
A fair assessment of the weaknesses of the study-population from one region and one center. Suggests planning a survey that takes into account:
- Very importantly, the % of data such as race, education level, family history of cancer and the impact of the first line of treatment is decreasing with subsequent years.
The data ends with 2016-I suggest doing another similar work with current data 2017-2023 to see if it continues to look worse and worse over the years will take into account data from a larger part of the country
Response: We have included the following paragraph in the conclusion. Thank you so much for this sugeestion. We really appreciated it !
« We hope to continue this research, analyzing the subsequent time series (2017-2023) as soon as the data are available, aiming to compare the trends in completeness of the variables present in the tumor file for better monitoring ».
Please add that the education data is not only very important because of the late reporting of cancer to the doctor. They are also important because of the later consequences of lack of cardiooncologic knowledge: survivors of childhood cancer are at high risk of oncologic background complications in the future. Hence the need for regular examinations in adulthood. Please quote:
Leszek P, Klotzka A, Bartuś S, Burchardt P, Czarnecka AM, Długosz-Danecka M, et al. A practical approach to the 2022 ESC cardio-oncology guidelines: Comments by a team of experts - cardiologists and oncologists. Kardiol Pol. 2023;81(10):1047-1063. doi: 10.33963/v.kp.96840.
Response: Thank you so Much for this suggestion.
We have included the following paragraph and added the recommended reference.
Data regarding education are not important only due to the late notification of cancer by the oncologist. Its relevance also extends to the subsequent implications of the absence of oncological knowledge. For instance, survivors of childhood cancer face a high risk of cardio-oncological complications in the future and, given this, it is imperative to carry out regular exams in adulthood [34].
- Add to the conclusions that the proposal to improve the number of data. Perhaps if the assignment of appropriate treatment depended on the completion of this data, the data would be more complete?
Response: Thank you for this suggestion. We have included the following paragraph in the conclusion to addressing it:
A strategy to mitigate the occurrence of unfilled information would be the implementation of continuous training for registrars, highlighting the relevance of these reistries as a basis for planning care, monitoring patients and quality of care provided in hospital institutions as well as for conducting research focused on cancer surveillance.
Round 2
Reviewer 1 Report
Comments and Suggestions for Authors
Line 102. CNS without capital letters.
Author Response
REVIEWER 1 - Decision: (Manuscript Accepted)
Comments and Suggestions for Authors
Line 102. CNS without capital letters.
Response: Ok. Done! Thanks for all the comments and for your contributions to our manuscript !
Reviewer 2 Report
Comments and Suggestions for Authors
Please incorporate the suggestions in well defined manner. I am not satisfied with the modified manuscript. Hope to see more improvise version of this manuscript.
Comments on the Quality of English LanguageModerate English changes are required.
Author Response
REVIEWER 2
Authors have written a research article entitled “Temporal Trends in the Completeness of Epidemiological Variables in a Hospital-Based Cancer Registry of a Pediatric Oncology Center in Brazil”. The theme is interesting; however, the manuscript is not properly discussed in context to results and data/experimental findings was not properly represented that the authors should clarify before a new peer-revision round. The study design of the manuscript is not impressive and does not add some more information related to the impact of this kind of study on Pediatric Oncology. Therefore, the manuscript is not suitable for publication; it should be carefully revised before the following peer-review process. Response: We believe, as you said at the beginning of your comment, that The theme is interesting. Furthermore, this is a pioneering study in the only specialized and reference center in Pediatric Oncology in the Espírito Santo state, Brazil. The Discussion was expanded according to pertinent notes from the other 3 reviewers, all of whom have already accepted the manuscript in this new round and presented version.
What are the most recent and important achievements in the field related to present work? In my opinion, answers to these questions should be emphasized. Authors have not explained the rationale of the study to a greater extent. How could this study enhance the understanding/knowledge of researcher or clinicians by using this type of data. Response: Thank you for your comments. We have added some paragraphs in the discussion as well as conclusion. In conclusion, the completeness of most variables evaluated in the HEINSG's HbCR in Espírito Santo, Brazil, was rated as excellent. However, the variable race/color showed an increasing trend of incompleteness, while TNM displayed a downward trend between 2010 and 2016. To better understand the health-disease process, high-quality data in HbCR is necessary. A strategy to mitigate the occurrence of unfilled information would be the implementation of continuous training for registrars, highlighting the relevance of these registries as a basis for planning care, monitoring patients and quality of care provided in hospital institutions as well as for conducting research focused on cancer surveillance.
- Results presented in this study are very limited, all the experimental analysis based upon secondary data of cancer. These data could not lead to the conclusion of present study. Some more parameters must be added in this study. Please use reference: https://doi.org/10.1016/j.canep.2022.102242
Response: Reference yours added as per suggested. Thanks! “A recent study which aimed to investigate temporal trends and factors associated with the cancer diagnosed at stage IV in Brazil in two decades, have showed that there was a high frequency of cancer diagnosed at stage IV and an increasing trend in different cancer types, which were associated with distinct sociodemographic, lifestyle, and clinical factors [36].”
In addition, we have added your comment in the limitation section. Thanks!
“Second, all the analysis reported are based upon secondary data of cancer, and as we know, inherent to all health information systems there are missing and incomplete data”
Finally, he present study draws attention to the importance and relevance of complete and reliable data on HbCRs, When data follows the principles of accuracy, completeness, conformity, punctuality, consistency and integrity, some actions become even more efficient, for example, accurate and reliable decision-making by healthcare professionals and clinicians can be guaranteed by good management of data quality.
- There is poor presentation of data like very less use of graph and figure.
Response: We have added one more Figure.
- The discussion should be rather organized around arguments avoiding simply describing details without providing much meaning. Authors have discussed some content and then seems not interested in writing more content which makes discussion part incomplete.
Response: Thank you for this comment. The Discussion was expanded according to pertinent notes from the other 3 reviewers who already endorsed our manuscript for publication.
Reviewer 3 Report
Comments and Suggestions for Authors
Review comments on previous version have been satisfactorily adressed.
Author Response
REVIEWER 3 - Decision: (Manuscript Accepted)
Comments and Suggestions for Authors
Review comments on previous version have been satisfactorily adressed.
Response: Thank you soi much for all the suggestions, recommendations as well as your contributions to our manuscript.